# The Influence of Parental Sleep and Experiences Related to COVID-19 on Sleep in Children and Adolescents between 2020 and 2021 in Brazil

**DOI:** 10.3390/ijerph20032638

**Published:** 2023-02-01

**Authors:** Samanta Andresa Richter, Luísa Basso Schilling, Clarissa Ferraz-Rodrigues, Nathália Fritsch Camargo, Magda Lahorgue Nunes

**Affiliations:** 1Ph.D. Program on Pediatrics and Child Health, Pontifical Catholic University of Rio Grande do Sul (PUCRS), Porto Alegre 90619-900, Brazil; 2School of Medicine, Pontifical Catholic University of Rio Grande do Sul (PUCRS), Porto Alegre 90619-900, Brazil; 3School of Health and Life Sciences, Pontifical Catholic University of Rio Grande do Sul (PUCRS), Porto Alegre 90619-900, Brazil; 4Brain Institute—Task Force COVID-19, Porto Alegre 90619-900, Brazil

**Keywords:** sleep, parents, children, adolescents, COVID-19

## Abstract

The study aimed to evaluate the influence of parental sleep and experiences related to COVID-19 on sleep changes in children and adolescents in the period between 2020 and 2021 in Brazil and further compare the sleep of adults with and without children. This is a longitudinal web-survey study. Participants were invited to respond to a questionnaire regarding sleep characteristics, mental health issues, and work/lifestyle modifications in two waves of the pandemic (April–July 2020 and 2021). A total of 1172 adults answered both questionnaires, and 281 were dyads (parent–child/adolescent). Parent and non-parent adult responders had similar sociodemographic data, with a predominance of the female sex and self-declared white skin color prevailing along with higher levels of education in both groups. The prevalence of sleep problems in adults varied from 20.6% to 30.2% in the parent group and from 16.9% to 30.1% in non-parents. The prevalence of sleep problems in children and adolescents raised from 2020 to 2021 (respectively, 48% and 49.5%) but differences were not statistically significant. The multivariate logistic model showed in both years that changes in children’s/adolescents’ sleep was related to parents working at home, infected family/friends, time of exposure to COVID-19, and daytime sleep dysfunction. Our data showed that parental sleep and experiences related to COVID-19 influenced sleep changes in children and adolescents. Parents had a significant difference in daytime sleepiness compared to the group without children.

## 1. Introduction

At the end of December 2019, new cases of a severe acute pneumonia of unknown origin appeared in Wuhan City, Hubei Province, China. The newly discovered virus belonging to the *Coronaviridae* family became known as SARS-CoV-2 and quickly spread across all continents [1]. In March 2020, the World Health Organization (WHO) declared a state of public calamity for the pandemic. Characterized by its high transmissibility, the disease caused by the virus was called COVID-19 [1]. More than two years later, it is estimated that the worldwide mortality associated with the COVID-19 pandemic was 14.9 million between 2020 and 2021 [2].

In order to contain the rapid spread of SARS-CoV-2, measures such as the use of protective masks and social isolation started worldwide. In Brazil, social isolation due to COVID-19 started in the second week of March 2020. Only essential services remained open. The abrupt closure caused schools to have their operations interrupted and adapted to the online model [3].

The unexpected and prolonged interruption of normal school routines, daily activities, and social connections resulted in decreased general well-being, increased stress levels, and changes in sleep patterns [3], immediately affecting the mental health status of the population [4].

Good quality of sleep is essential for physical and mental health [5]. Significant changes in routine during isolation, including more flexible sleep schedules, poorer sleep quality, prolonged daytime naps, increased screen exposure, decreased daylight exposure, reduced physical activity, and increased sedentary behavior, as well as increased stress and anxiety, contributed to the consolidation of unhealthy sleep patterns and increased prevalence of sleep disorders [6]. The National Sleep Foundation coined the term “Coronasomnia” to refer to sleep changes related to the pandemic [7]. Through the changes imposed by COVID-19 and based on parental sleep, this study aimed to evaluate the influence of parental sleep and experiences related to COVID-19 on sleep changes in children and adolescents in the period between 2020 and 2021 in Brazil.

## 2. Materials and Methods

This is a longitudinal nationwide web-survey study conducted by the COVID-19 Task Force of the Brain Institute (BraIns) of the Pontifical Catholic University of Rio Grande do Sul, Brazil. Data were collected using the Qualtrics^®^ online survey software platform (www.qualtrics.com). Participants were invited through the “virtual snowball” sampling method which began by sending invitations through social networks and internet, radio, and TV in two periods, the first between 27 April and 30 July 2020, comprising the first wave of the COVID-19 pandemic in Brazil, and the second period from 5 April to 30 July 2021. The model of social isolation imposed by COVID-19 varied during data collection, where in 2020, Brazil was in complete lockdown, and in 2021, social isolation was partial.

After a brief explanation regarding the aim of the study, participants who agreed to join clicked on the option “I agree to participate in this study” for online informed consent and were then directed to a sociodemographic questionnaire and questions related to COVID-19. Afterward, they were directed to answer questionnaires related to sleep, and participants who had children under 18 years old answered sleep questionnaires regarding their children. All the responses were anonymous and without any other identification of the participants. However, we had asked to leave an e-mail contact if the responder was interested in being surveyed in the second phase of the study. The information was directly stored in the Qualtrics platform and was available only to the senior researcher under the use of a password.

The national survey had a total of 5007 respondents in 2020 and 1582 in 2021, of these, 1172 participated in both years and constitute the sample analyzed in this work. The final sample was divided into adults with children (parents) and non-parents. The present study was approved by the Institutional Ethics Committee and registered on Plataforma Brasil under number 30748320.5.0000.5336.

### 2.1. Instruments

The evaluation of sleep during the survey was based on previously validated questionnaires. The Brazilian version of the “Pittsburgh Sleep Quality Index (PSQI)” was applied to parents and adolescents (13 to 17 years and 11 months). It has seven components that together lead to a global score. In the global score, a score equal to or greater than 5 distinguishes between poor and good sleep quality, and above 10 points indicates a predisposition to sleep disorders [8]. The Brazilian version of the “Epworth Sleepiness Scale (ESS)” was used only in adults. The ESS is an eight-item survey where participants score, on a Likert scale of 0–3, the probability of napping or falling asleep in life activities daily. Out of a total of 24 points, a score > 10 indicates mild or excessive daytime sleepiness [9].

The “Brief Infant Sleep Questionnaire (BISQ)” was used for children aged 0 to 3 years and 11 months. For the BISQ, the cutoff score was one (or more) of the following: more than three nocturnal awakenings, nighttime wakefulness greater than one hour, and/or total sleep time less than nine hours [10].

The “Sleep Disorders Scale for Children (SDSC)” was used for children aged 4 to 12 years and 11 months. The SDSC scores were calculated as follows: sleep initiation and maintenance disorders (sum of items 1, 2, 3, 4, 5, 10, and 11, cut-off point 17); sleep breathing disorders (sum of items 13, 14, and 15, cut-off point 7; arousal disorders (sum of items 17, 20, and 21, cut-off point 6); position disorders (sum of items 6, 7, 8, 12, 18, and 19, cut-off point 14 points); excessive daytime sleepiness (sum of items 22, 23, 24, 25, and 26, cut-off point 13); sleep hyperhidrosis (sum of items 9 and 16, cut-off point 7). The cutoff of 40 points from the total score was used [11].

### 2.2. Comparison between Parents and Non-Parents

The PSQI and ESS scales were compared between the groups (parents x non-parents) in each year of investigation. We sought to identify whether the observed proportions of sleep problems and each of the components of the scales differed between the groups, as well as in each year of the evaluation.

### 2.3. Statistical Analysis

Data were analyzed using the Statistical Package for Social Sciences version 20.0 (SPSS Inc., Chicago, IL, USA, 2008) for Windows, with a significance level of 5% for statistical decision criteria. Results were presented using descriptive statistics—absolute and relative distribution (*n*—%), as well as mean, standard deviation, and amplitude. The study of age symmetry was carried out using the Shapiro–Wilk test. The analysis that compared the paired proportions between the two years of investigation was the McNemar–Bowker test. Since, in the evaluation of the impact observed between the proportions of each year, the size of the effect was calculated through Cramer’s V, considering the classification criterion: <0.10 = insignificant effect; 0.10–0.39 small effect; 0.50–0.69 = average effect; and >0.70 = large effect [12]. In the comparisons between independent groups (gender, age group, and mental health), Pearson’s chi-square test (Monte Carlo Simulation) was used.

### 2.4. Logistic Regression

The analysis that involved the prediction of sleep alteration in children and adolescents was investigated using Binary Multiple Logistic Regression. In the selection of independent variables with predictive potential to characterize sleep disorders in children, the Backward conditional method was used and, to verify the goodness of fit of the final logistic regression model, the Nagelkerk and Hosmer–Lemeshow R^2^ estimators were considered. The probability of the gradual entry of the variables in the model was used as 0.05 and for the removal of 0.10. Thus, the independent variables in the model were analyzed by the characteristics of the parents who presented a minimum level of significance less than or equal to (*p* ≤ 0.250) in relation to the child’s sleep disorder. In addition, the sex and age range of the children were considered in the model.

## 3. Results

### 3.1. Sociodemographic Data

The results presented refer to a sample of 281 parents and 891 adult non-parents investigated in 2020 and 2021, with parental ages ranging from 19 to 51 years (37.3 ± 11.8) and 18 to 91 (38.6 ± 13.9), respectively. The female sex and self-declared white skin color prevailed in the two groups with higher levels of an academic degree. In relation to monthly income, the highest concentrations occurred in Classes A and C. Considering sociodemographic data, no significant differences were observed between the groups. Our sample included responders from all Brazilian regions; however, the largest concentration was from the southern region of the country (76.1%) (Table 1).

### 3.2. Labor Characterization and Relationship with COVID-19

In Table 2, we described labor characterization and different aspects related to COVID-19; again, no significant differences were observed between the groups. Many responders were health professionals, followed by teachers. Regarding the parents infected, most of them had mild to moderate symptoms (25.3%). Considering issues related to mental health, more than half of the sample reported feeling alone, mentioned that their lives had no meaning, that they self-harmed themselves for some reason, or even had suicidal ideation during the pandemic period.

### 3.3. Parents’ and Non-Parents’ Sleep

The PSQI and ESS scales were compared between the groups in each year of investigation. Significant differences were observed in PSQI items, such as sleep quality and efficiency. In the subjective component of sleep quality, in 2020, parents claimed worse sleep quality, and in 2021, poor quality was higher in the non-parent group. Regarding sleep efficiency, which was considered normal if >85%, in 2020, parents reported higher efficiency than non-parents. In 2021, there was an inversion in this result, with a decrease in sleep efficiency for those who had children (Table 3).

Regarding the results referring to the ESS scale, significant differences were observed in the items “watching TV”, “sitting quietly in a public place”, and “Sitting quietly after a lunch without alcohol” (Table 4). In 2020, it was found that parents were more likely to fall asleep watching TV than in 2021 and for those who did not have children. In 2021, parents were more likely to sleep when sitting and quiet in a public place (*p* < 0.001). Regarding the total score of ESS, in both groups there was an increase from 2020 to 2021, meaning that diurnal somnolence increased in the second year of the pandemic (*p* < 0.001). As a result, it was noted that in 2020, adults napped less, regardless of whether they were parents or not. However, in 2021 parents had a lower proportion of naps when compared to non-parents.

### 3.4. Sleep of Children and Adolescents

Among children/adolescents included, there was a predominance of the male sex (54.1%) and the majority in the age range between 4 and 12 years (47.3%). In all age ranges, the prevalence of sleep problems was high according to the instruments used, in both years, but without significant differences from 2020 to 2021. However, when parents had sleep disturbances or when children, adolescents, and parents both had sleep disturbances, it proved to be an important aspect in the quality of sleep of children and adolescents (Table 5).

### 3.5. Logistic Regression

Table 6 shows the factors associated with the presence of sleep disorders in children and adolescents in 2020 and 2021 using the multivariate logistic model. In 2020, potential predictors of sleep disturbances in children and adolescents were parents working fully from home, parents having been infected by COVID-19, and parents’ daytime sleep dysfunction. Adolescents had a higher risk of disorders of sleep compared to the age group of 0 to 3 years. In 2021, predictors of sleep disorders in children and adolescents were related to parents who needed to be exposed to COVID-19 for more than 12 h a day and any family member/close friend who had been infected; parents with sleep efficiency below 85% were more likely to have children with sleep disorders. Children aged 4 to 12 years were more likely to have sleep disorders when compared to those aged 0 to 3 years.

## 4. Discussion

In this longitudinal study, it was possible to show that sleep problems persisted in all age ranges across the pandemic. In 2020, both parents and adults who did not have children showed changes in sleep, mainly daytime sleepiness that few said they previously had, but in 2021, daytime sleepiness had a considerable increase. In 2020, adolescents had the highest percentage of sleep disorders, and in 2021, it was school-age children. Regarding aspects related to COVID-19, contagion from close relatives and the exposure time of adults influenced the sleep of children and adolescents.

We know that during the social isolation caused by the COVID-19 pandemic, lifestyle, work, and habits/routines were suddenly changed [13]. Parents found themselves on a mission to work at home and adapt the family to the new routine. In addition, anxiety, fear of the unknown, and economic recession interacted to increase sleep complaints [14]. Even at the beginning of the pandemic, studies showed that parents had difficulty sleeping due to their routines that were also abruptly changed and new family arrangements because of the lockdown [15,16].

It was observed in our study that children and adolescents whose parents worked full-time at home were more predisposed to sleep disorders. This finding corroborates an Italian study carried out with mothers who were working at home, which observed a worsening in the quality of sleep and distortion of the experience of time in mothers and children, in addition to an increase in emotional symptoms and difficulties in self-regulation in children [16].

Studies involving adolescents have shown that sleep disturbances during the COVID-19 pandemic have been associated with increased insomnia, anxiety, and depression, in addition to increased rates of psychiatric symptoms. Loneliness due to social isolation can put children and adolescents at risk for the appearance of psychiatric disorders [17,18,19]. Our study did not explore the emotional aspects of children and adolescents, but adults were asked about aspects of mental health and it was found that more than half of the sample had moments when they felt alone during the first year of the pandemic. Corroborating our study, a multicenter study carried out in 87 countries showed that during the COVID-19 pandemic, negative emotions increased worldwide, potentiating mental illness [20].

As for the contamination of parents by COVID-19 and family members or friends, it influenced the worsening of sleep in children and adolescents. These issues may be linked to feelings experienced, such as the fear of losing a loved one. A study carried out in Canada showed that in 2020, changes in family sleep habits were associated with fears and concerns about COVID-19; in addition, reduced access to health services was also associated with parental concerns [21]. It is important to note that in our study, most respondents were health professionals, so the possibility of exposure to the virus was greater, represented by 40.2% of parents and 22.8% of non-parents. Many studies have associated sleep with the emotional aspects of health professionals, but the literature is scarce when it is focused on the children of these professionals [20,22,23].

As for the limitations of the study, the asymmetric regional distribution of participants with most of the sample from southern Brazil should be mentioned. However, in this region, lockdown protocols were more homogeneous than in other regions of the country. Another limiting aspect is the higher prevalence of respondents with an academic degree, which may make the generalization of the findings to the whole population difficult. As it is known, the pandemic has had different impacts on people from different socioeconomic levels. In addition, the questionnaire allowed parents to respond in relation to only one child, losing control or guaranteeing which criteria the parents used to choose the child they were responding to (the worst sleeper or the youngest, etc.).

## 5. Conclusions

Our data showed that during the COVID-19 pandemic, parental sleep and lifestyle issues influenced sleep alterations in children and adolescents across the years 2020 and 2021. Furthermore, it is important to highlight the persistently high rates of sleep disorders in all age groups over the two years, especially in school-age children and adolescents. When comparing sleep characteristics between the two adult study groups, parents had a worse sleep quality.

## Figures and Tables

**Table 1 ijerph-20-02638-t001:** Characterization of the socio-demographic profile of adult respondents in 2020 and 2021.

Variables	Parents (*n* = 281) ^A^		No Parents (*n* = 891) ^A^	
	*n*	%	*n*	%
**Age (years)**	37.3 ± 11.8 (19–51)		38.6 ± 13.9 (18–91)	
**Female sex**	222	79.1	667	74.9
**Color**				
White	233	82.9	764	85.8
Brown	37	13.2	79	8.9
Black	7	2.5	29	3.5
Others (yellow, indigenous, and without declaration)	4	1.4	18	2.1
**Education**				
Elementary School to High School	12	4.3	44	4.9
University graduate	57	20.3	426	47.8
Postgraduate	212	75.5	421	47.2
**Income ***				
Class A	148	52.7	272	30.5
Class B	14	5.0	129	14.5
Class C	76	27.1	371	41.6
Class D	14	5.0	81	9.1
Class E	4	1.4	38	4.3
**Country region**				
South	189	67.3	703	78.9
Southeast	60	21.4	128	14.4
Midwest	7	2.5	15	1.7
Northeast	19	6.8	40	4.5
North	6	2.1	5	0.6

^A^ = Percentages obtained based on the total sample; * According to the Instituto Brasileiro de Geografia e Estatística (IBGE 2020) approximate values in US dollars: Class A (>US 2250); Class B (US 1730 to 2250); Class C (US 400 to 1729); Class D (US 250 to 400); Class E ≤ US 250.

**Table 2 ijerph-20-02638-t002:** Labor characterization of adults’ responders and relationship with COVID-19.

Variables	Parents (*n* = 281) ^A^		No Parents (*n* = 891) ^A^	
	*n*	%	*n*	*%*
**Working currently (yes)**	252	89.7	644	72.3
**Professional category**				
Retired	-	-	4	0.5
Autonomous	17	6.7	59	9.2
Commerce	6	2.4	33	3.7
Student	2	0.8	58	9.1
Teacher	57	22.7	115	17.9
Health professional working on the frontline/COVID-19	48	19.1	63	9.8
Health professional not working on the frontline/COVID-19	65	25.8	140	21.8
Others	86	34.1	418	64.9
**Occupation directly exposes you to the possibility of being infected with COVID-19 (yes)**	178	70.6	408	63.4
**Daily exposure to COVID-19**				
Less than 4 h	71	28.2	166	25.8
From 5 to 6 h	36	14.3	95	14.8
From 6 to 12 h	59	23.4	130	20.2
More than 12 h	12	4.8	17	2.7
**Way of working**				
Home office and workplace	68	27.0	175	27.2
Full home office	75	29.8	287	44.6
Fully on-site (no reduction in hours)	88	34.9	180	28.0
Reduced hours	19	7.5	39	6.1
**Isolation in the city (2021)**				
Lockdown	25	8.9	61	6.8
Partially	245	87.2	789	88.6
Normal	11	3.9	41	4.6
**How long has the city been in this type of isolation**				
±15 days	58	20.6	217	24.4
Between 16 to 30 days	76	27.1	251	28.2
Between 31 to 60 days	71	25.3	202	22.7
>60 days	76	27.1	221	24.8
**Was contaminated by COVID-19 during the pandemic (yes)**	71	25.3	159	17.8
**Symptoms**				
Asymptomatic	12	16.9	21	13.2
Light or moderate	56	78.9	130	81.8
Severe	3	4.2	8	5.1
**Family member or close friend who became infected with COVID-19 (yes)**	257	91.5	785	88.1
**Family member or close friend who passed away from COVID-19 (yes)**	140	49.8	348	39.1
**Mental Health**				
In the last year, have you felt lonely (yes)	158	56.3	591	66.3
In the last year, have you felt like your life is meaningless (yes)	94	33.5	419	47.1
During the past year, have you ever inflicted self-harm (yes)	120	42.7	359	40.3
During this time of the pandemic, have you had or are having suicidal thoughts (yes)	26	9.3	170	19.1
In the last year, did you ever try to take your own life (yes)	1	0.4	10	1.1
In the last year, have you witnessed an episode of violence with someone who lives with you (yes)	27	9.7	62	7.0
**Vaccine**				
Have you been vaccinated? * (yes)	122	43.4	269	30.2
**Doses**				
One dose	56	20.0	132	14.8
Two doses	66	23.5	137	15.4

^A^: Percentages obtained based on the total sample; * Questionnaire available between 5 April and 31 July 2021. In Brazil, vaccination started on 17 January 2021 for risk groups.

**Table 3 ijerph-20-02638-t003:** Adult absolute and relative distribution for sleep component ratings by year of assessment based on PSQI.

PSQI	Year of Investigation and Children (Yes × No)												
	2020 ^A^					2021 ^A^					Significance Levels		
	Parents		No Parents		Δ ^D^	Parents		No Parents		Δ ^D^	2020	2021	Δ ^G^
	*n*	%	*n*	*%*		*n*	%	*n*	%				
**Sleep quality**											0.119	0.288	0.036
Very good	116	41.3	349	39.2	−2.1	19	6.8	70	7.9	1.1			
Fairly good	86	30.6	327	36.7	6.1	131	46.6	402	45.1	−1.5			
Fairly bad	78	27.8	205	23.0	−4.8	98	34.9	344	38.6	3.7			
Very bad	1	0.4	10	1.1	0.8	33	11.7	75	8.4	−3.3			
**Sleep latency**											0.395	0.419	0.122
≤15 min	74	26.3	251	28.2	1.8	38	13.5	105	11.8	−1.7			
From 16 to 30 min	133	47.3	400	44.9	−2.4	88	31.3	280	31.4	0.1			
From 31 to 60 min	56	19.9	201	22.6	2.6	76	27.0	283	31.8	4.7			
>60 min	18	6.4	39	4.4	−2.0	79	28.1	223	25.0	−3.1			
**Sleep duration**											0.346	0.795	0.076
>7 h	51	18.1	165	18.5	0.4	80	28.5	239	26.8	−1.6			
From 6:01 to 7:00 h	103	36.7	346	38.8	2.2	110	39.1	375	42.1	2.9			
From 5:00 to 6:00 h	124	44.1	357	40.1	−4.1	70	24.9	220	24.7	−0.2			
<5 h	3	1.1	23	2.6	1.5	21	7.5	57	6.4	−1.1			
**Sleep efficiency**											0.316	0.450	0.109
≥85%	71	25.3	192	21.5	−3.8	42	14.9	111	12.5	−2.5			
<85%	210	74.7	699	78.5	3.8	239	85.1	780	87.5	2.5			
**Difficulty falling asleep**											0.654	0.701	0.511
Not during the past month	68	24.2	185	20.8	−3.4	3	1.1	9	1.0	−0.1			
Less than once a week	124	44.1	419	47.0	2.9	146	52.0	450	50.5	−1.5			
Once or twice a week	65	23.1	213	23.9	0.8	120	42.7	377	42.3	−0.4			
Three or more times a week	24	8.5	74	8.3	−0.2	12	4.3	55	6.2	1.9			
**Use of sleeping medication**											0.404	0.417	0.288
Not during the past month	201	71.5	622	69.8	−1.7	206	73.3	637	71.5	−1.8			
Less than once a week	19	6.8	89	10.0	3.2	20	7.1	93	10.4	3.3			
Once or twice a week	44	15.7	135	15.2	−0.5	15	5.3	47	5.3	−0.1			
Three or more times a week	17	6.0	45	5.1	−1.0	40	14.2	114	12.8	−1.4			
**Daytime dysfunction**											0.902	0.927	0.203
Not during the past month	66	23.5	206	23.1	−0.4	27	9.6	87	9.8	0.2			
Less than once a week	192	68.3	613	68.8	0.5	139	49.5	446	50.1	0.6			
Once or twice a week	19	6.8	54	6.1	−0.7	85	30.2	275	30.9	0.6			
Three or more times a week	4	1.4	18	2.0	0.6	30	10.7	83	9.3	−1.4			
**Global score PSQI**											0.346	0.657	0.104
Good Sleep Quality Global Score < 5	83	29.5	264	29.6	0.1	43	15.3	118	13.2	−2.1			
Poor Sleep Quality Global Score ≥ 5	140	49.8	476	53.4	3.6	153	54.4	505	56.7	2.2			
Sleep Disorder Predisposition > 10	58	20.6	151	16.9	−3.7	85	30.2	268	30.1	−0.2			

^A^: Percentages obtained based on the total number of valid cases and each year; Δ: difference between the proportions of the group without children in relation to the group with children in each classification category. ^D^: Pearson’s chi-square test comparing parent and non-parent groups. ^G^: Pearson’s chi-square test comparing the proportions of differences (Δ) between the years 2020 and 2021.

**Table 4 ijerph-20-02638-t004:** General characterization of the adult sample by year of assessment based on the Epworth Sleepiness Scale (ESS).

ESS	Year of Investigation and Children (Yes × No)												
	2020 ^A^					2021 ^A^					Significance Levels		
	Parents		No Parents		Δ ^D^	Parents		No Parents		Δ ^D^	2020	2021	Δ ^G^
	*n*	%	*n*	%		*n*	%	*n*	%				
**Sitting and Reading**											0.184	0.610	0.156
No chance of dozing	194	69.0	628	70.5	1.4	42	14.9	159	17.8	2.9			
Slight chance of dozing	85	30.2	243	27.3	−3.0	96	34.2	314	35.2	1.1			
Moderate chance of dozing	2	0.7	20	2.2	1.5	83	29.5	245	27.5	−2.0			
High chance of dozing	0	0.0	0	0.0	-	60	21.4	173	19.4	−1.9			
**Watching TV**											0.857	0.319	0.026
No chance of dozing	199	70.8	619	69.5	−1.3	29	10.3	121	13.6	3.3			
Slight chance of dozing	81	28.8	267	30.0	1.1	96	34.2	299	33.6	−0.6			
Moderate chance of dozing	1	0.4	5	0.6	0.2	91	32.4	249	27.9	−4.4			
High chance of dozing					-	65	23.1	222	24.9	1.8			
**Sitting inactive in a public place (e.g., a theater or a meeting)**											0.077	0.718	<0.001 *
No chance of dozing	200	71.2	655	73.5	2.3	136	48.4	423	47.5	−0.9			
Slight chance of dozing	79	28.1	213	23.9	−4.2	89	31.7	308	34.6	2.9			
Moderate chance of dozing	2	0.7	23	2.6	1.9	39	13.9	105	11.8	−2.1			
High chance of dozing	0	0.0	0	0.0	-	17	23.6	55	6.2	−17.4 *			
**As a passenger in a car for an hour without a break**											0.858	0.057	0.033
No chance of dozing	196	69.8	622	69.8	0.1	107	38.1	307	34.5	−3.6			
Slight chance of dozing	84	29.9	263	29.5	−0.4	75	26.7	295	33.1	6.4 *			
Moderate chance of dozing	1	0.4	6	0.7	0.3	57	20.3	136	15.3	−5.0			
High chance of dozing					-	42	14.9	153	17.2	2.2			
**Lying down to rest in the afternoon when** **circumstances permit**											0.149	0.424	0.011
No chance of dozing	204	72.6	599	67.2	−5.4 *	182	64.8	601	67.5	2.7			
Slight chance of dozing	75	26.7	276	31.0	4.3 *	99	35.2	290	32.5	−2.7			
Moderate chance of dozing	2	0.7	16	1.8	1.1	0	0	0.0	0	0.0			
**Sitting and talking to someone**											0.976	0.405	0.021
No chance of dozing	196	69.8	624	70.0	0.3	182	64.8	601	67.5	2.7			
Slight chance of dozing	84	29.9	263	29.5	−0.4	99	35.2	290	32.5	−2.7			
Moderate chance of dozing	1	0.4	4	0.4	0.1	0	0.0	0	0.0	-			
**Sitting quietly after a lunch without alcohol**											0.926	0.232	<0.001 *
No chance of dozing	195	69.4	624	70.0	0.6	45	16.0	145	16.3	0.3			
Slight chance of dozing	85	30.2	263	29.5	−0.7	76	27.0	295	33.1	6.1 *			
Moderate chance of dozing	1	0.4	4	0.4	0.1	78	27.8	228	25.6	−2.2			
High chance of dozing	0	0.0	0	0.0	-	82	29.2	223	25.0	−4.2			
**In a car, while stopped for a few minutes in traffic**											0.697	0.918	0.274
No chance of dozing	200	71.2	641	71.9	0.8	207	73.7	654	73.4	−0.3			
Slight chance of dozing	81	28.8	248	27.8	−1.0	54	19.2	178	20.0	0.8			
Moderate chance of dozing	0	0.0	2	0.2	0.2	14	5.0	37	4.2	−0.8			
High chance of dozing	0	0.0	0	0.0	-	6	2.1	22	2.5	0.3			
**Total Score ESS**											0.328	0.056	<0.001 *
Lower Normal Daytime Sleepiness	269	95.7	828	92.9	−2.8	92	32.7	344	38.6	5.9 *			
Higher Normal Daytime Sleepiness	11	3.9	60	6.7	2.8	104	37.0	308	34.6	−2.4			
Mild Excessive Daytime Sleepiness	1	0.4	2	0.2	−0.1	30	10.7	94	10.5	−0.1			
Moderate Excessive Daytime Sleepiness	0	0.0	1	0.1	0.1	42	14.9	86	9.7	−5.3 *			
Severe Excessive Daytime Sleepiness	0	0.0	0	0.0	-	13	4.6	59	6.6	2.0			

^A^: Percentages obtained based on the total number of valid cases and each group (with and without children); Δ variation between the proportions with children vs. no children (* significant differences in comparisons between the two years). ^D^: Pearson’s chi-square test. ^G^: Pearson’s chi-square test comparing the proportions of differences (Δ) between the years 2020 and 2021.

**Table 5 ijerph-20-02638-t005:** Sleep characterization of children and adolescent respondents in 2020 and 2021.

Variables	Year of Investigation					
	2020 (*n* = 281)		2021 (*n* = 281)		Difference	*p*
	n	%	n	%		
**Sex**						
Male	152	54.1	152	54.1		
Female	129	45.9	129	45.9		
**Age**						
From 0 to 3 years and 11 months	84	29.9	65	23.1		
From 4 years to 12 years and 11 months	118	42.0	133	47.3		
From 13 years to 17 years and 11 months	79	28.1	83	29.5		
**Lives with parents**						
Yes	259	92.2	262	93.2		
No	22	7.8	19	6.8		
**Type of childbirth**						
Natural	71	25.3	71	25.3		
Caesarian	210	74.7	210	74.7		
**Birth complications**						
Yes	32	11.4	32	11.4		
No	249	88.6	249	88.6		
**Screen time**						
Does not use	8	2.8	-	-		
<1 h	35	12.5	36	12.8		
>1 h to 3 h	71	25.3	121	43.1		
>3 h to 5 h	115	40.9	89	31.6		
>5 h	52	18.5	35	12.5		
**Education**						
Preschool	75	26.7	65	23.1		
Elementary School	127	45.2	141	50.2		
High school	70	24.9	75	26.7		
Do not study	9	3.2	-	-		
**0–3 years and 11 months (*n* = 65)**						
**Abnormal BISQ**						
More than three nocturnal awakenings	8	30.8	7	28.0		
Nocturnal wakefulness	16	61.5	13	52.0		
Total sleep times < 9 h	19	73.1	15	60.0		
**4–12 years and 11 months (*n* = 133)**						
**Abnormal SDSC**	61	45.9	71	53.4		
Disorders of initiating and maintaining sleep	20	32.8	24	33.8		
Sleep breathing disorders	1	1.6	1	1.4		
Disorders of arousal	7	11.5	8	11.3		
Sleep–wake transition disorders	8	13.1	9	12.7		
Excessive daytime sleepiness	4	6.6	5	7.0		
Sleep hyperhidrosis	5	8.2	6	8.5		
**13–17 years and 11 months (*n* = 83)**						
Poor Sleep Quality (PSQI Global score ≥ 5)	48	57.8	43	51.8		
**Child with sleep disturbance ***						
Yes	135	48.0	139	49.5	−1.5	0.975
No	146	52.0	142	50.5	1.5	
**Parents with sleep disturbance ****						
Yes	198	70.5	238	84.7	41.0	
No	83	29.5	43	15.3	69.4	<0.001 ***
**Child and parents with sleep disturbance (2020 *n* = 135) (2021 *n* = 139)**						
Yes	98	72.6	115	82.7	45.2	<0.001 ***
No	37	27.4	24	17.3	65.4	

OBS: BISQ = Brief infant sleep questionnaire, SDSC = Sleep disturbance scale for children, PSQI = Pittsburg sleep quality index; * estimated by BISQ (0–3 years and 11 months), SDSC (4–12 years and 11 months) and PSQI (13–17 years and 11 months) scales. ** Adults estimated by the PSQI scale (>5 points). *** statistical significance.

**Table 6 ijerph-20-02638-t006:** Factors associated with sleep alterations in children and adolescents in 2020–2021 through the multivariate logistic model.

Independent Variables 2020 (*n* = 281)	Sig.	Odds Ratio	IC95% Odds Ratio		Independent Variables 2021 (*n* = 281)	Sig.	Odds Ratio	IC95% Odds Ratio	
			Lower	Higher				Lower	Higher
**Way of working**					**Daily exposure to COVID-19**	0.038			
Home office and workplace	0.044	1.000	-	-	Less than 4 h		2.449	0.592	10.123
Full home office		2.786	1.351	5.746	From 5 to 6 h		2.301	0.537	9.858
Fully on-site (no reduction in hours)		1.841	0.927	3.656	From 6 to 12 h		2.630	0.573	12.061
Reduced hours		2.801	0.937	8.375	More than 12 h		6.241	1.421	27.406
**Was contaminated by COVID-19 during the pandemic (yes)**	0.026	1.963	1.086	3.547	**F** **amily member or close friend who became infected with COVID-19 (yes)**	0.002	3.658	1.333	10.035
**PSQI—Sleep efficiency**	0.031				**PSQI—Sleep efficiency**	0.026			
<85%		1.000	-	-	>85%		1.000	-	-
>85%		0.720	0.208	2.491	<85%		2.311	1.011	6.590
**PSQI—Daytime dysfunction**	0.011				**PSQI—Daytime dysfunction**	0.017			
Not during the past month		1.000	-	-	Not during the past month		1.0	-	-
Less than once a week		1.024	0.516	2.031	Less than once a week		1.893	0.522	6.859
Once or twice a week		2.528	1.019	12.219	Once or twice a week		1.452	1.108	2.427
Three or more times a week		1.240	1.017	3.323	Three or more times a week		1.679	1.126	3.721
**Age of children and adolescents**	0.044				**Age of children and adolescents**	0.028			
From 0 to 3 years and 11 months		1.0	-	-	From 0 to 3 years and 11 months		1.000	-	-
From 4 years to 12 years and 11 months		1.428	0.754	2.705	From 4 years to 12 years and 11 months		2.271	1.147	4.497
From 13 years to 17 years and 11 months		2.334	1.149	4.741	From 13 years to 17 years and 11 months		1.832	0.871	3.855

Regression model parameters. Final model: Nagelkerk’s R^2^ = 0.377; Cox & Nel = 0.302; 2LL = 355.395; Hosmer–Lemeshow test (Chi square) (8) = 8.916; *p* = 0.349; Confusion matrix: Total 74.8%.

## Data Availability

Not applicable.

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
