# Peer review of "The Influence of Parental Sleep and Experiences Related to COVID-19 on Sleep in Children and Adolescents between 2020 and 2021 in Brazil"

_ijerph, 2023, doi:10.3390/ijerph20032638_

Round 1
Reviewer 1 Report
This study aims to evaluate the influence of parental sleep during COVID-19 pandemia in children and adolescents. To do so, participants responded to questionnaires regarding sleep characteristics, mental Health issues and work/lifestyle modifications in 2020 and 2021. Although the number of participants is adequate, results are confusing and discussion and conclusions did not match with the results. Methodology also needs to be clarified.
- The objectives should be clearly described in the introduction and results should be written in accordance.
- Since the comparison between 2021 and 2020 is not objective of the work and also taking into account that the percentage of adults with sleep problems did not differ between 2020 and 2021, there is no point in separating the two years. I suggest to carry the analysis considering the two years together. This would increase the n and improve the results.
- Since authors want to evaluate the influence of parental sleep in children’s sleep, associations (correlations) between parents responses and responses of their kids should be studied. Also, questions such as “Which is the percentage of children with sleep alteration that have parents with sleep alterations? Or which is the percentage of children with sleep problems with parents working full time at home?” should be addressed.
- Tables 1 and 2. There is a large description of participants characterization which is not relevant for the study. On the other hand, information about children characteristics (attending school, sex,...) is missing. Table 2 could be presented as supplementary information.
- Table 3 refers to sleep alterations in adults, however, since the objective of the study is to evaluate children’s sleep, responses to PSQI or to other sleep questionnaires of children should also be specified.
- Table 3: Please indicate if bold characters refer to statistical significant differences. Significance levels >0.05 are bold marked. Please check.
- Tables 3 and 4: Bonferroni or any test for multiple comparisons should be applied to determine the statistically significant p value.
- Table 6. Authors carried out a multiple logistic regression to determine the factors associated with sleep alterations in children. Which criteria have been used to select the independent variables to be entered in the analysis? The factors before and after the backwards selection (first and final model) should be indicated together with their significance. Table 6 should also indicate the number of participants.
- Abstract should clearly express the results. Line24. “The prevalence of sleep problems in children and adolescents is high..:” Please specify the level of “high”. Line 25: what do authors exactly refer to as “increased risk”?
Author Response
To the Editor and Reviewer 1
Dear all,
We were pleased to receive a feedback on our manuscript and we thank reviewer 1 for his/her work.
We replied to all comments to and please find amended text section in red.
We hope the new version will be considered satisfactory for publication in the journal and we look forward to receiving news from you.
Reviewer 1
- The objectives should be clearly described in the introduction and results should be written in accordance.
We adjusted the objective of the study, where we want to compare the years 2020 and 2021.
- Since the comparison between 2021 and 2020 is not objective of the work and also taking into account that the percentage of adults with sleep problems did not differ between 2020 and 2021, there is no point in separating the two years. I suggest to carry the analysis considering the two years together. This would increase the n and improve the results.
We believe that with the adjustment of the objective of the study, it is important to maintain the comparison between 2020 and 2021, it is noteworthy that in the ESS scale we had statistical differences between the years.
- Since authors want to evaluate the influence of parental sleep in children’s sleep, associations (correlations) between parents responses and responses of their kids should be studied. Also, questions such as “Which is the percentage of children with sleep alteration that have parents with sleep alterations? Or which is the percentage of children with sleep problems with parents working full time at home?” should be addressed.
We include this information in Table 5.
- Tables 1 and 2. There is a large description of participants characterization which is not relevant for the study. On the other hand, information about children characteristics (attending school, sex,...) is missing. Table 2 could be presented as supplementary information.
As we have compared adults with children and without we do think that it is important to keep this information.
On table 5 we show results regarding the children/adolescent population.
- Table 3 refers to sleep alterations in adults, however, since the objective of the study is to evaluate children’s sleep, responses to PSQI or to other sleep questionnaires of children should also be specified.
All information on children and adolescent sleep is available on tables 5 and 6.
-
3: Please indicate if bold characters refer to statistical significant differences. Significance levels >0.05 are bold marked. Please check.
We adjusted Table 3 because the bold does not represent significant statistics, there was a formatting error.
- Tables 3 and 4: Bonferroni or any test for multiple comparisons should be applied to determine the statistically significant p value.
We verified that the multiple comparisons test was not applied, we did not compare mean scores, in fact we compared proportions by Person's Chi-Square test to define the statistically significant p.
- Table 6. Authors carried out a multiple logistic regression to determine the factors associated with sleep alterations in children. Which criteria have been used to select the independent variables to be entered in the analysis? The factors before and after the backwards selection (first and final model) should be indicated together with their significance. Table 6 should also indicate the number of participants.
For logistic regression, we used the characteristics of parents who had a minimum level of significance less than or equal to 0.250 (p ≤ 0.250) as the choice of independent variables, and also considered the gender and age of the children. This explanation is highlighted in the text. The number of participants has been entered.
- Abstract should clearly express the results. Line24. “The prevalence of sleep problems in children and adolescents is high..:” Please specify the level of “high”. Line 25: what do authors exactly refer to as “increased risk”?
We have rephrased the information.

Reviewer 2 Report
Remarks:
page 2, line 55
“…the term 55 “Coronasomnia,” to refer to sleep challenges related to the pandemic.” sleep challenges -> sleep changes ???
page 2, line 57-58
“…the influence of parents' sleep and experiences related to COVID-19 on sleep changes in children and adolescents…”
At this point, the Authors should clarify the rationale for this hypothesis, i.e. how do parents' sleep and experiences affect their children's sleep? What could be the mechanism of this phenomenon?
As to ‘sleep changes’ – do you mean before/during pandemic or 2020/2021 differences?
page 2, l.69
replace comma with point/full stop.
page 2, line 91
“The ESS is a 6-item survey…” ESS in its original version had 8 items. Please explain.
page 4, line 158 and Table 2
“…they punished themselves for some reason or even had suicidal ideation during the pandemic period”; “During the past year, have you ever punished yourself for a situation?”
Do you really mean ‘punishment’ and not ‘blame’?
page 7, Table 3. - Sleep latency
De 16 30 minutes
De 31a 60 minutes
– please translate ;)
page 7, Table 3. - Sleep duration
From 6 to 7 hours
From 5 to 6 hours
So where/in which category do you put 6 hours exactly?
page 11, line 204
Factors associated with sleep alterations in children and adolescents…
alterations –> presence of sleep disorders?
page 13, line 210
“…sleep alterations have persisted in all age ranges across the pandemic.”
Maybe ‘sleep problems’? ‘Alterations’ suggest that you had also some baseline data, i.e. before pandemic.
page 13, line 211
“Both adults’ groups showed changes mainly in daytime sleepiness, where in 2020 few claimed to have, but in 2021 daytime sleepiness had a considerable increase. In the first year, adolescents had the highest percentage of sleep alterations and in 2021, were school-age children” – Please improve the style.
page 13, line 215 contamination -> infection, contagion?
page 13, line 219
“In addition, anxiety, fear of the unknown and economic recession collaborated to increase sleep complaints”
collaborated -> interacted
page 13, line 240
“Regarding the contamination of parents by COVID-19 and family members or friends…” – style!
Author Response
To the Editor and Reviewer 2
Dear all,
We were pleased to receive a feedback on our manuscript and we thank reviewer 2 for his/her work.
We replied to all comments to and please find amended text section in red.
We hope the new version will be considered satisfactory for publication in the journal and we look forward to receiving news from you.
Reviewer 2
page 2, line 55
“…the term 55 “Coronasomnia,” to refer to sleep challenges related to the pandemic.” sleep challenges -> sleep changes ???
We correct the term.
page 2, line 57-58
“…the influence of parents' sleep and experiences related to COVID-19 on sleep changes in children and adolescents…”
At this point, the Authors should clarify the rationale for this hypothesis, i.e. how do parents' sleep and experiences affect their children's sleep? What could be the mechanism of this phenomenon?
As to ‘sleep changes’ – do you mean before/during pandemic or 2020/2021 differences?
We entered the rationale and considered sleep changes between the years 2020 to 2021.
page 2, l.69
replace comma with point/full stop.
We change the comma to full stop.
page 2, line 91
“The ESS is a 6-item survey…” ESS in its original version had 8 items. Please explain.
We checked and there was a typo, 8 items of the ESS Scale were checked in the study.
page 4, line 158 and Table 2
“…they punished themselves for some reason or even had suicidal ideation during the pandemic period”; “During the past year, have you ever punished yourself for a situation?”
Do you really mean ‘punishment’ and not ‘blame’?
We refer to the act of self-harm, so we changed it to make it clearer to the reader.
page 7, Table 3. - Sleep latency
De 16 30 minutes
De 31a 60 minutes
– please translate ;)
Sorry for the mistake, we have adjusted to English.
page 7, Table 3. - Sleep duration
From 6 to 7 hours
From 5 to 6 hours
So where/in which category do you put 6 hours exactly?
We adjust from 5:00 to 6:00 hours and from 6:01 to 7:00 hours.
page 11, line 204
Factors associated with sleep alterations in children and adolescents…
alterations –> presence of sleep disorders?
We adjust the term.
page 13, line 210
“…sleep alterations have persisted in all age ranges across the pandemic.”
Maybe ‘sleep problems’? ‘Alterations’ suggest that you had also some baseline data, i.e. before pandemic.
We adjust the term.
page 13, line 211
“Both adults’ groups showed changes mainly in daytime sleepiness, where in 2020 few claimed to have, but in 2021 daytime sleepiness had a considerable increase. In the first year, adolescents had the highest percentage of sleep alterations and in 2021, were school-age children” – Please improve the style.
We rewrote the paragraph adjusting the style.
page 13, line 215 contamination -> infection, contagion?
We adjust the term.
page 13, line 219
“In addition, anxiety, fear of the unknown and economic recession collaborated to increase sleep complaints”
collaborated -> interacted
We adjust the term.
page 13, line 240
“Regarding the contamination of parents by COVID-19 and family members or friends…” – style!
We adjust the style.
